# Probabilistic low-rank matrix completion on finite alphabets

**Jean Lafond**
Institut Mines-Télécom
Télécom ParisTech
CNRS LTCI
jean.lafond@telecom-paristech.fr

**Olga Klopp**
CREST et MODAL'X
Université Paris Ouest
Olga.KLOPP@math.cnrs.fr

**Éric Moulines**
Institut Mines-Télécom
Télécom ParisTech
CNRS LTCI
moulines@telecom-paristech.fr

**Joseph Salmon**
Institut Mines-Télécom
Télécom ParisTech
CNRS LTCI
joseph.salmon@telecom-paristech.fr

## Abstract

The task of reconstructing a matrix given a sample of observed entries is known as the *matrix completion problem*. It arises in a wide range of problems, including recommender systems, collaborative filtering, dimensionality reduction, image processing, quantum physics or multi-class classification to name a few. Most works have focused on recovering an unknown real-valued low-rank matrix from randomly sub-sampling its entries. Here, we investigate the case where the observations take a finite number of values, corresponding for examples to ratings in recommender systems or labels in multi-class classification. We also consider a general sampling scheme (not necessarily uniform) over the matrix entries. The performance of a nuclear-norm penalized estimator is analyzed theoretically. More precisely, we derive bounds for the Kullback-Leibler divergence between the true and estimated distributions. In practice, we have also proposed an efficient algorithm based on lifted coordinate gradient descent in order to tackle potentially high dimensional settings.

## 1 Introduction

Matrix completion has attracted a lot of contributions over the past decade. It consists in recovering the entries of a potentially high dimensional matrix, based on their random and partial observations. In the classical noisy matrix completion problem, the entries are assumed to be real valued and observed in presence of additive (homoscedastic) noise. In this paper, it is assumed that the entries take values in a finite alphabet that can model categorical data. Such a problem arises in analysis of voting patterns, recovery of incomplete survey data (typical survey responses are true/false, yes/no or do not know, agree/disagree/indifferent), quantum state tomography [13] (binary outcomes), recommender systems [18, 2] (for instance in common movie rating datasets, *e.g.,* MovieLens or Neflix, ratings range from 1 to 5) among many others. It is customary in this framework that rows represent individuals while columns represent items *e.g.,* movies, survey responses, etc. Of course, the observations are typically incomplete, in the sense that a significant proportion of the entries are missing. Then, a crucial question to be answered is whether it is possible to predict the missing entries from these partial observations.

Since the problem of matrix completion is ill-posed in general, it is necessary to impose a low-dimensional structure on the matrix, one particularly popular example being a low rank constraint. The classical noisy matrix completion problem (real valued observations and additive noise), can be solved provided that the unknown matrix is low rank, either exactly or approximately; see [7, 15, 17, 20, 5, 16] and the references therein. Most commonly used methods amount to solve a least square program under a rank constraint or a convex relaxation of a rank constraint provided by the nuclear (or trace norm) [10].

The problem of probabilistic low rank matrix completion over a finite alphabet has received much less attention; see [22, 8, 6] among others. To the best of our knowledge, only the binary case (also referred to as the 1-bit matrix completion problem) has been covered in depth. In [8], the authors proposed to model the entries as Bernoulli random variables whose success rate depend upon the matrix to be recovered through a convex link function (logistic and probit functions being natural examples). The estimated matrix is then obtained as a solution of a maximization of the log-likelihood of the observations under an explicit low-rank constraint. Moreover, the sampling model proposed in [8] assumes that the entries are sampled uniformly at random. Unfortunately, this condition is not totally realistic in recommender system applications: in such a context some users are more active than others and some popular items are rated more frequently. Theoretically, an important issue is that the method from [8] requires the knowledge of an upper bound on the nuclear norm or on the rank of the unknown matrix.

Variations on the 1-bit matrix completion was further considered in [6] where a max-norm (though the name is similar, this is different from the sup-norm) constrained minimization is considered. The method of [6] allows more general non-uniform samplings but still requires an upper bound on the max-norm of the unknown matrix.

In the present paper we consider a penalized maximum log-likelihood method, in which the log-likelihood of the observations is penalized by the nuclear norm (*i.e.,* we focus on the Lagrangian version rather than on the constrained one). We first establish an upper bound of the Kullback-Leibler divergence between the true and the estimated distribution under general sampling distributions; see Section 2 for details. One should note that our method only requires the knowledge of an upper bound on the maximum absolute value of the probabilities, and improves upon previous results found in the literature.

Last but not least, we propose an efficient implementation of our statistical procedure, which is adapted from the lifted coordinate descent algorithm recently introduced in [9, 14]. Unlike other methods, this iterative algorithm is designed to solve the convex optimization and not (possibly non-convex) approximated formulation as in [21]. It also has the benefit that it does not need to perform full/partial SVD (Singular Value Decomposition) at every iteration; see Section 3 for details.

**Notation**

Define $m_1 \wedge m_2 := \min(m_1, m_2)$ and $m_1 \vee m_2 := \max(m_1, m_2)$. We equip the set of $m_1 \times m_2$ matrices with real entries (denoted $\mathbb{R}^{m_1 \times m_2}$) with the scalar product $\langle X | X' \rangle := \mathrm{tr}(X^\top X')$. For a given matrix $X \in \mathbb{R}^{m_1 \times m_2}$ we write $\|X\|_\infty := \max_{i,j} |X_{i,j}|$ and, for $q \geq 1$, we denote its Schatten $q$-norm by

$$\|X\|_{\sigma,q} := \left( \sum_{i=1}^{m_1 \wedge m_2} \sigma_i(X)^q \right)^{1/q},$$

where $\sigma_i(X)$ are the singular values of $X$ ordered in decreasing order (see [1] for more details on such norms). The operator norm of $X$ is given by $\|X\|_{\sigma,\infty} := \sigma_1(X)$. Consider two vectors of $p-1$ matrices $(X^j)_{j=1}^{p-1}$ and $(X'^j)_{j=1}^{p-1}$ such that for any $(k,l) \in [m_1] \times [m_2]$ we have $X_{k,l}^j \geq 0$, $X_{k,l}^{'j} \geq 0$, $1 - \sum_{j=1}^{p-1} X_{k,l}^j \geq 0$ and $1 - \sum_{j=1}^{p-1} X_{k,l}^{'j} \geq 0$. Their square Hellinger distance is

$$d_H^2(X, X') := \frac{1}{m_1 m_2} \sum_{\substack{k \in [m_1] \\ l \in [m_2]}} \left[ \sum_{j=1}^{p-1} \left( \sqrt{X_{k,l}^j} - \sqrt{X_{k,l}^{'j}} \right)^2 + \left( \sqrt{1 - \sum_{j=1}^{p-1} X_{k,l}^j} - \sqrt{1 - \sum_{j=1}^{p-1} X_{k,l}^{'j}} \right)^2 \right]$$

and their Kullback-Leibler divergence is

$$\mathrm{KL}\left(X, X'\right) := \frac{1}{m_1 m_2} \sum_{\substack{k \in [m_1] \\ l \in [m_2]}} \left[ \sum_{j=1}^{p-1} X_{k,l}^j \log \frac{X_{k,l}^j}{X_{k,l}'^j} + (1 - \sum_{j=1}^{p-1} X_{k,l}^j) \log \frac{1 - \sum_{j=1}^{p-1} X_{k,l}^j}{1 - \sum_{j=1}^{p-1} X_{k,l}'^j} \right] .$$

Given an integer $p > 1$, a function $f : \mathbb{R}^{p-1} \to \mathbb{R}^{p-1}$ is called a $p$-link function if for any $x \in \mathbb{R}^{p-1}$ it satisfies $f^j(x) \geq 0$ for $j \in [p-1]$ and $1 - \sum_{j=1}^{p-1} f^j(x) \geq 0$. For any collection of $p-1$ matrices $(X^j)_{j=1}^{p-1}$, $f(X)$ denotes the vector of matrices $(f(X)^j)_{j=1}^{p-1}$ such that $f(X)_{k,l}^j = f(X_{k,l}^j)$ for any $(k,l) \in [m_1] \times [m_2]$ and $j \in [p-1]$.

## 2   Main results

Let $p$ denote the cardinality of our finite alphabet, that is the number of classes of the logistic model (*e.g.*, ratings have $p$ possible values or surveys $p$ possible answers). For a vector of $p-1$ matrices $X = (X^j)_{j=1}^{p-1}$ of $\mathbb{R}^{m_1 \times m_2}$ and an index $\omega \in [m_1] \times [m_2]$, we denote by $X_\omega$ the vector $(X_\omega^j)_{j=1}^{p-1}$. We consider an *i.i.d.* sequence $(\omega_i)_{1 \leq i \leq n}$ over $[m_1] \times [m_2]$, with a probability distribution function $\Pi$ that controls the way the matrix entries are revealed. It is customary to consider the simple uniform sampling distribution over the set $[m_1] \times [m_2]$, though more general sampling schemes could be considered as well. We observe $n$ independent random elements $(Y_i)_{1 \leq i \leq n} \in [p]^n$. The observations $(Y_1, \ldots, Y_n)$ are assumed to be independent and to follow a multinomial distribution with success probabilities given by

$$\mathbb{P}(Y_i = j) = f^j(\bar{X}_{\omega_i}^1, \ldots, \bar{X}_{\omega_i}^{p-1}) \quad j \in [p-1] \quad \text{and} \quad \mathbb{P}(Y_i = p) = 1 - \sum_{j=1}^{p-1} \mathbb{P}(Y_i = j)$$

where $\{f^j\}_{j=1}^{p-1}$ is a $p$-link function and $\bar{X} = (\bar{X}^j)_{j=1}^{p-1}$ is the vector of true (unknown) parameters we aim at recovering. For ease of notation, we often write $\bar{X}_i$ instead of $\bar{X}_{\omega_i}$. Let us denote by $\Phi_Y$ the (normalized) negative log-likelihood of the observations:

$$\Phi_Y(X) = -\frac{1}{n} \sum_{i=1}^{n} \left[ \sum_{j=1}^{p-1} \mathbb{1}_{\{Y_i=j\}} \log \left( f^j(X_i) \right) + \mathbb{1}_{\{Y_i=p\}} \log \left( 1 - \sum_{j=1}^{p-1} f^j(X_i) \right) \right] , \quad (1)$$

For any $\gamma > 0$ our proposed estimator is the following:

$$\hat{X} = \underset{\substack{X \in (\mathbb{R}^{m_1 \times m_2})^{p-1} \\ \max_{j \in [p-1]} \|X^j\|_\infty \leq \gamma}}{\arg\min} \Phi_Y^\lambda(X) , \quad \text{where} \quad \Phi_Y^\lambda(X) = \Phi_Y(X) + \lambda \sum_{j=1}^{p-1} \|X^j\|_{\sigma,1} , \quad (2)$$

with $\lambda > 0$ being a regularization parameter controlling the rank of the estimator. In the rest of the paper we assume that the negative log-likelihood $\Phi_Y$ is convex (this is the case for the multinomial logit function, see for instance [3]).

In this section we present two results controlling the estimation error of $\hat{X}$ in the binomial setting (*i.e.,* when $p = 2$). Before doing so, let us introduce some additional notation and assumptions. The score function (defined as the gradient of the negative log-likelihood) taken at the true parameter $\bar{X}$, is denoted by $\bar{\Sigma} := \nabla \Phi_Y(\bar{X})$. We also need the following constants depending on the link function $f$ and $\gamma > 0$:

$$M_\gamma = \sup_{|x| \leq \gamma} 2|\log(f(x))| ,$$

$$L_\gamma = \max \left( \sup_{|x| \leq \gamma} \frac{|f'(x)|}{f(x)}, \sup_{|x| \leq \gamma} \frac{|f'(x)|}{1 - f(x)} \right) ,$$

$$K_\gamma = \inf_{|x| \leq \gamma} \frac{f'(x)^2}{8 f(x)(1 - f(x))} .$$

In our framework, we allow for a general distribution for observing the coefficients. However, we need to control deviations of the sampling mechanism from the uniform distribution and therefore we consider the following assumptions.

**H1.** *There exists a constant $\mu \geq 1$ such that for all indexes $(k,l) \in [m_1] \times [m_2]$*

$$\min_{k,l}(\pi_{k,l}) \geq 1/(\mu m_1 m_2) \ .$$

*with $\pi_{k,l} := \Pi(\omega_1 = (k,l))$.*

Let us define $C_l := \sum_{k=1}^{m_1} \pi_{k,l}$ (resp. $R_k := \sum_{l=1}^{m_2} \pi_{k,l}$) for any $l \in [m_2]$ (resp. $k \in [m_1]$) the probability of sampling a coefficient in column $l$ (resp. in row $k$).

**H2.** *There exists a constant $\nu \geq 1$ such that*

$$\max_{k,l}(R_k, C_l) \leq \nu/(m_1 \wedge m_2) \ ,$$

Assumption H1 ensures that each coefficient has a non-zero probability of being sampled whereas H2 requires that no column nor row is sampled with too high probability (see also [11, 16] for more details on this condition).

We define the sequence of matrices $(E_i)_{i=1}^{n}$ associated to the revealed coefficient $(\omega_i)_{i=1}^{n}$ by $E_i := e_{k_i}(e'_{l_i})^{\top}$ where $(k_i, l_i) = \omega_i$ and with $(e_k)_{k=1}^{m_1}$ (*resp.* $(e'_l)_{l=1}^{m_2}$) being the canonical basis of $\mathbb{R}^{m_1}$ (*resp.* $\mathbb{R}^{m_2}$). Furthermore, if $(\varepsilon_i)_{1 \leq i \leq n}$ is a Rademacher sequence independent from $(\omega_i)_{i=1}^{n}$ and $(Y_i)_{1 \leq i \leq n}$ we define

$$\Sigma_R := \frac{1}{n} \sum_{i=1}^{n} \varepsilon_i E_i \ .$$

We can now state our first result. For completeness, the proofs can be found in the supplementary material.

**Theorem 1.** *Assume H1 holds, $\lambda \geq 2\|\bar{\Sigma}\|_{\sigma,\infty}$ and $\|\bar{X}\|_\infty \leq \gamma$. Then, with probability at least $1 - 2/d$ the Kullback-Leibler divergence between the true and estimated distribution is bounded by*

$$\mathrm{KL}\left(f(\bar{X}), f(\hat{X})\right) \leq 8 \max\left(\frac{\mu^2}{K_\gamma} m_1 m_2 \operatorname{rank}(\bar{X}) \left(\lambda^2 + c^* L_\gamma^2 (\mathbb{E}\|\Sigma_R\|_{\sigma,\infty})^2\right), \mu e M_\gamma \frac{\sqrt{\log(d)}}{n}\right),$$

*where $c^*$ is a universal constant.*

Note that $\|\bar{\Sigma}\|_{\sigma,\infty}$ is stochastic and that its expectation $\mathbb{E}\|\Sigma_R\|_{\sigma,\infty}$ is unknown. However, thanks to Assumption H2 these quantities can be controlled.

To ease notation let us also define $m := m_1 \wedge m_2$, $M := m_1 \vee m_2$ and $d := m_1 + m_2$.

**Theorem 2.** *Assume H1 and H2 hold and that $\|\bar{X}\|_\infty \leq \gamma$. Assume in addition that $n \geq 2m \log(d)/(9\nu)$. Taking $\lambda = 6L_\gamma\sqrt{2\nu \log(d)/(mn)}$, then with probability at least $1 - 3/d$ the folllowing holds*

$$K_\gamma \frac{\|\bar{X} - \hat{X}\|_{\sigma,2}^2}{m_1 m_2} \leq \mathrm{KL}\left(f(\bar{X}), f(\hat{X})\right) \leq \max\left(\bar{c} \frac{\nu \mu^2 L_\gamma^2}{K_\gamma} \frac{M \operatorname{rank}(\bar{X}) \log(d)}{n}, 8\mu e M_\gamma \frac{\sqrt{\log(d)}}{n}\right),$$

*where $\bar{c}$ is a universal constant.*

*Remark.* Let us compare the rate of convergence of Theorem 2 with those obtained in previous works on 1-bit matrix completion. In [8], the parameter $\bar{X}$ is estimated by minimizing the negative log-likelihood under the constraints $\|X\|_\infty \leq \gamma$ and $\|X\|_{\sigma,1} \leq \gamma\sqrt{rm_1 m_2}$ for some $r > 0$. Under the assumption that $\operatorname{rank}(\bar{X}) \leq r$, they could prove that

$$\frac{\|\bar{X} - \hat{X}\|_{\sigma,2}^2}{m_1 m_2} \leq C_\gamma \sqrt{\frac{rd}{n}} \ ,$$

where $C_\gamma$ is a constant depending on $\gamma$ (see [8, Theorem 1]). This rate of convergence is slower than the rate of convergence given by Theorem 2. [6] studied a max-norm constrained maximum likelihood estimate and obtained a rate of convergence similar to [8].

# 3 Numerical Experiments

**Implementation** For numerical experiments, data were simulated according to a multinomial logit distribution. In this setting, an observation $Y_{k,l}$ associated to row $k$ and column $l$ is distributed as $\mathbb{P}(Y_{k,l} = j) = f^j(X_{k,l}^1, \ldots, X_{k,l}^{p-1})$ where

$$f^j(x_1, \ldots, x_{p-1}) = \exp(x_j)\left(1 + \sum_{j=1}^{p-1} \exp(x_j)\right)^{-1}, \quad \text{for } j \in [p-1] . \tag{3}$$

With this choice, $\Phi_Y$ is convex and problem (2) can be solved using convex optimization algorithms. Moreover, following the advice of [8] we considered the unconstrained version of problem (2) (*i.e.,* with no constraint on $\|X\|_\infty$), which reduces significantly the computation burden and has no significant impact on the solution in practice. To solve this problem, we have extended to the multinomial case the coordinate gradient descent algorithm introduced by [9]. This type of algorithm has the advantage, say over the Soft-Impute [19] or the SVT [4] algorithm, that it does not require the computation of a full SVD at each step of the main loop of an iterative (proximal) algorithm (bare in mind that the proximal operator associated to the nuclear norm is the soft-thresholding operator of the singular values). The proposed version only computes the largest singular vectors and singular values. This potentially decreases the computation by a factor close to the value of the upper bound on the rank commonly used (see the aforementioned paper for more details).

Let us present the algorithm. Any vector of $p-1$ matrices $X = (X^j)_{j=1}^{p-1}$ is identified as an element of the tensor product space $\mathbb{R}^{m_1 \times m_2} \otimes \mathbb{R}^{p-1}$ and denoted by:

$$X = \sum_{j=1}^{p-1} X^j \otimes e^j , \tag{4}$$

where again $(e^j)_{j=1}^{p-1}$ is the canonical basis on $\mathbb{R}^{p-1}$ and $\otimes$ stands for the tensor product. The set of normalized rank-one matrices is denoted by

$$\mathcal{M} := \left\{ M \in \mathbb{R}^{m_1 \times m_2} | M = uv^\top \mid \|u\| = \|v\| = 1, u \in \mathbb{R}^{m_1}, v \in \mathbb{R}^{m_2} \right\} .$$

Define $\Theta$ the linear space of real-valued functions on $\mathcal{M}$ with finite support, *i.e.,* $\theta(M) = 0$ except for a finite number of $M \in \mathcal{M}$. This space is equipped with the $\ell^1$-norm $\|\theta\|_1 = \sum_{M \in \mathcal{M}} |\theta(M)|$. Define by $\Theta_+$ the positive orthant, *i.e.,* the cone of functions $\theta \in \Theta$ such that $\theta(M) \geq 0$ for all $M \in \mathcal{M}$. Any tensor $X$ can be associated with a vector $\theta = (\theta^1, \ldots, \theta^{p-1}) \in \Theta_+^{p-1}$, *i.e.,*

$$X = \sum_{j=1}^{p-1} \sum_{M \in \mathcal{M}} \theta^j(M) M \otimes e^j . \tag{5}$$

Such representations are not unique, and among them, the one associated to the SVD plays a key role, as we will see below. For a given $X$ represented by (4) and for any $j \in \{1, \ldots, p-1\}$, denote by $\{\sigma_k^j\}_{k=1}^{n^j}$ the (non-zero) singular values of the matrix $X^j$ and $\{u_k^j, v_k^j\}_{k=1}^{n^j}$ the associated singular vectors. Then, $X$ may be expressed as

$$X = \sum_{j=1}^{p-1} \sum_{k=1}^{n^j} \sigma_k^j u_k^j (v_k^j)^\top \otimes e^j . \tag{6}$$

Defining $\theta^j$ the function $\theta^j(M) = \sigma_k^j$ if $M = u_k^j(v_k^j)^\top$, $k \in [n^j]$ and $\theta^j(M) = 0$ otherwise, one obtains a representation of the type given in Eq. (5).

Conversely, for any $\theta = (\theta^1, \ldots, \theta^{p-1}) \in \Theta^{p-1}$, define the map

$$W : \theta \to W_\theta := \sum_{j=1}^{p-1} W_\theta^j \otimes e^j \quad \text{with} \quad W_\theta^j := \sum_{M \in \mathcal{M}} \theta^j(M) M$$

and the auxiliary objective function

$$\tilde{\Phi}_Y^\lambda(\theta) = \lambda \sum_{j=1}^{p-1} \sum_{M \in \mathcal{M}} \theta^j(M) + \Phi_Y(W_\theta) . \tag{7}$$

The map $\theta \mapsto W_\theta$ is a continuous linear map from $(\Theta^{p-1}, \|\cdot\|_1)$ to $\mathbb{R}^{m_1 \times m_2} \otimes \mathbb{R}^{p-1}$, where $\|\theta\|_1 = \sum_{j=1}^{p-1} \sum_{M \in \mathcal{M}} |\theta^j(M)|$. In addition, for all $\theta \in \Theta_+^{p-1}$

$$\sum_{j=1}^{p-1} \|W_\theta^j\|_{\sigma,1} \leq \|\theta\|_1 ,$$

and one obtains $\|\theta\|_1 = \sum_{j=1}^{p-1} \|W_\theta^j\|_{\sigma,1}$ when $\theta$ is the representation associated to the SVD decomposition. An important consequence, outlined in [9, Proposition 3.1], is that the minimization of (7) is actually equivalent to the minimization of (2); see [9, Theorem 3.2].

The proposed coordinate gradient descent algorithm updates at each step the nonnegative finite support function $\theta$. For $\theta \in \Theta$ we denote by $\text{supp}(\theta)$ the support of $\theta$ and for $M \in \mathcal{M}$, by $\delta_M \in \Theta$ the Dirac function on $\mathcal{M}$ satisfying $\delta_M(M) = 1$ and $\delta_M(M') = 0$ if $M' \neq M$. In our experiments we have set to zero the initial $\theta_0$.

---

**Algorithm 1:** Multinomial lifted coordinate gradient descent

**Data**: Observations: $Y$, tuning parameter $\lambda$
initial parameter: $\theta_0 \in \Theta_+^{p-1}$; tolerance: $\epsilon$; maximum number of iterations: $K$
**Result**: $\theta \in \Theta_+^{p-1}$
**Initialization:** $\theta \leftarrow \theta_0$, $k \leftarrow 0$
**while** $k \leq K$ **do**

    **for** $j = 0$ *to* $p-1$ **do**
        Compute top singular vectors pair of $(-\nabla \Phi_Y(W_\theta))_j$: $u_j, v_j$

    Let $g = \lambda + \min_{j=1,\ldots,p-1} \langle \nabla \Phi_Y \mid u^j(v^j)^\top \rangle$
    **if** $g \leq -\epsilon/2$ **then**

        $(\beta_0, \ldots, \beta_{p-1}) = \underset{(b_0,\ldots,b_{p-1}) \in \mathbb{R}_+^{p-1}}{\arg\min} \tilde{\Phi}_Y^\lambda \left( \theta + (b_0 \delta_{u^0(v^0)^\top}, \ldots, b_{p-1} \delta_{u^{p-1}(v^{p-1})^\top}) \right)$
        $\theta \leftarrow \theta + (\beta_0 \delta_{u^0(v^0)^\top}, \ldots, \beta_{p-1} \delta_{u^{p-1}(v^{p-1})^\top})$
        $k \leftarrow k + 1$

    **else**

        Let $g_{\max} = \max_{j \in [p-1]} \max_{u^j(v^j)^\top \in \text{supp}(\theta^j)} |\lambda + \langle \nabla \Phi_Y \mid u^j(v^j)^\top \rangle|$
        **if** $g_{\max} \leq \epsilon$ **then**
            **break**

        **else**

            $\theta \leftarrow \underset{\theta' \in \Theta_+^{p-1}, \text{supp}(\theta'^j) \subset \text{supp}(\theta^j), j \in [p-1]}{\arg\min} \tilde{\Phi}_Y^\lambda(\theta')$
            $k \leftarrow k + 1$

---

A major interest of Algorithm 1 is that it requires to store the value of the parameter entries only for the indexes which are actually observed. Since in practice the number of observations is much smaller than the total number of coefficients $m_1 m_2$, this algorithm is both memory and computationally efficient. Moreover, using an SVD algorithm such as Arnoldi iterations to compute the top singular values and vector pairs (see [12, Section 10.5] for instance) allows us to take full advantage of gradient sparse structure. Algorithm 1 was implemented in C and Table 1 gives a rough idea of the execution time for the case of two classes on a 3.07Ghz w3550 Xeon CPU (RAM 1.66 Go, Cache 8Mo).

**Simulated experiments** To evaluate our procedure we have performed simulations for matrices with $p = 2$ or $5$. For each class matrix $X^j$ we sampled uniformly five unitary vector pairs $(u_k^j, v_k^j)_{k=1}^5$. We have then generated matrices of rank equals to 5, such that

$$X^j = \Gamma \sqrt{m_1 m_2} \sum_{k=1}^5 \alpha_k u_k^j (v_k^j)^\top ,$$

with $(\alpha_1, \ldots, \alpha_5) = (2, 1, 0.5, 0.25, 0.1)$ and $\Gamma$ is a scaling factor. The $\sqrt{m_1 m_2}$ factor, guarantees that $\mathbb{E}[\|X^j\|_\infty]$ does not depend on the sizes of the problem $m_1$ and $m_2$.

| Parameter Size | $10^3 \times 10^3$ | $3 \cdot 10^3 \times 3 \cdot 10^3$ | $10^4 \times 10^4$ |
|---|---|---|---|
| **Observations** | $10^5$ | $10^5$ | $10^7$ |
| **Execution Time (s.)** | 4.5 | 52 | 730 |

Table 1: Execution time of the proposed algorithm for the binary case.

We then sampled the entries uniformly and the observations according to a logit distribution given by Eq. (3). We have then considered and compared the two following estimators both computed using Algorithm 1:

- the logit version of our method (with the link function given by Eq. (3))

- the Gaussian completion method (denoted by $\hat{X}^{\mathcal{N}}$), that consists in using the Gaussian log-likelihood instead of the multinomial in (2), *i.e.,* using a classical squared Frobenius norm (the implementation being adapted mutatis mutandis). Moreover an estimation of the standard deviation is obtained by the classical analysis of the residue.

Contrary to the logit version, the Gaussian matrix completion does not directly recover the probabilities of observing a rating. However, we can estimate this probability by the following quantity:

$$\mathbb{P}(\hat{X}^{\mathcal{N}}_{k,l} = j) = F_{\mathcal{N}(0,1)}(p_{j+1}) - F_{\mathcal{N}(0,1)}(p_j) \text{ with } p_j = \begin{cases} 0 & \text{if } j = 1 \text{ ,} \\ \frac{j - 0.5 - \hat{X}^{\mathcal{N}}_{k,l}}{\hat{\sigma}} & \text{if } 0 < j < p \\ 1 & \text{if } j = p \text{ ,} \end{cases}$$

where $F_{\mathcal{N}(0,1)}$ is the cdf of a zero-mean standard Gaussian random variable.

As we see on Figure 1, the logistic estimator outperforms the Gaussian for both cases $p = 2$ and $p = 5$ in terms of the Kullback-Leibler divergence. This was expected because the Gaussian model allows uniquely symmetric distributions with the same variance for all the ratings, which is not the case for logistic distributions. The choice of the $\lambda$ parameter has been set for both methods by performing 5-fold cross-validation on a geometric grid of size $0.8 \log(n)$.

Table 2 and Table 3 summarize the results obtained for a $900 \times 1350$ matrix respectively for $p = 2$ and $p = 5$. For both the binomial case $p = 2$ and the multinomial case $p = 5$, the logistic model slightly outperforms the Gaussian model. This is partly due to the fact that in the multinomial case, some ratings can have a multi-modal distribution. In such a case, the Gaussian model is unable to predict these ratings, because its distribution is necessarily centered around a single value and is not flexible enough. For instance consider the case of a rating distribution with high probability of seeing 1 or 5, low probability of getting 2, 3 and 4, where we observed both 1's and 5's. The estimator based on a Gaussian model will tend to center its distribution around 2.5 and therefore misses the bimodal shape of the distribution.

| Observations | $10 \cdot 10^3$ | $50 \cdot 10^3$ | $100 \cdot 10^3$ | $500 \cdot 10^3$ |
|---|---|---|---|---|
| **Gaussian prediction error** | 0.49 | 0.34 | 0.29 | 0.26 |
| **Logistic prediction error** | 0.42 | 0.30 | 0.27 | 0.24 |

Table 2: Prediction errors for a binomial (2 classes) underlying model, for a $900 \times 1350$ matrix.

| Observations | $10 \cdot 10^3$ | $50 \cdot 10^3$ | $100 \cdot 10^3$ | $500 \cdot 10^3$ |
|---|---|---|---|---|
| **Gaussian prediction error** | 0.78 | 0.76 | 0.73 | 0.69 |
| **Logistic prediction error** | 0.75 | 0.54 | 0.47 | 0.43 |

Table 3: Prediction Error for a multinomial (5 classes) distribution against a $900 \times 1350$ matrix.

**Real dataset**    We have also run the same estimators on the MovieLens $100k$ dataset. In the case of real data we cannot calculate the Kullback-Leibler divergence since no ground truth is available. Therefore, to compare the prediction errors, we randomly selected 20% of the entries as a test set, and the remaining entries were split between a training set (80%) and a validation set (20%).

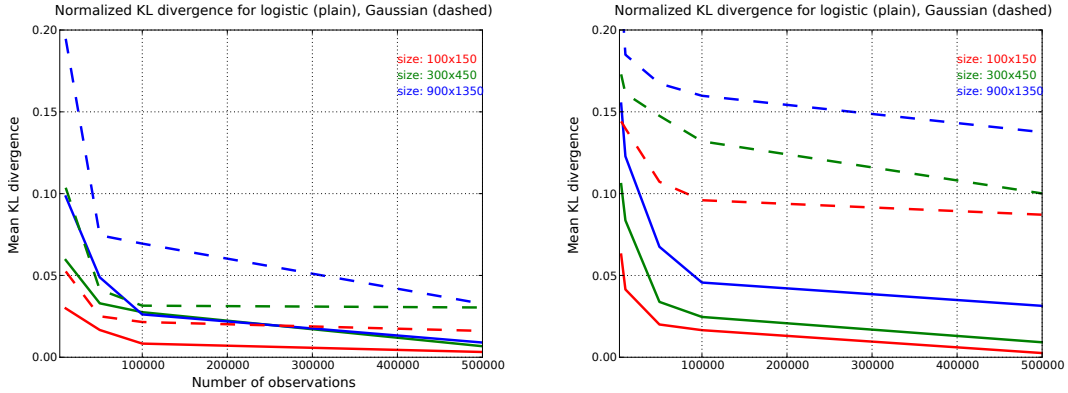

Figure 1: Kullback-Leibler divergence between the estimated and the true model for different matrices sizes and sampling fraction, normalized by number of classes. Right figure: binomial and Gaussian models ; left figure: multinomial with five classes and Gaussian model. Results are averaged over five samples.

For this dataset, ratings range from 1 to 5. To consider the benefit of a binomial model, we have tested each rating against the others (*e.g.,* ratings 5 are set to 0 and all others are set to 1). Interestingly we see that the Gaussian prediction error is significantly better when choosing labels $-1$, 1 instead of labels 0, 1. This is another motivation for not using the Gaussian version: the sensibility to the alphabet choice seems to be crucial for the Gaussian version, whereas the binomial/multinomial ones are insensitive to it. These results are summarized in table 4.

| Rating | 1 | 2 | 3 | 4 | 5 |
|---|---|---|---|---|---|
| **Gaussian prediction error (labels $-1$ and 1)** | 0.06 | 0.12 | 0.28 | 0.35 | 0.19 |
| **Gaussian prediction error (labels 0 and 1)** | 0.12 | 0.20 | 0.39 | 0.46 | 0.30 |
| **Logistic prediction error** | 0.06 | 0.11 | 0.27 | 0.34 | 0.20 |

Table 4: Binomial prediction error when performing one versus the others procedure on the Movie-Lens $100k$ dataset.

# 4 Conclusion and future work

We have proposed a new nuclear norm penalized maximum log-likelihood estimator and have provided strong theoretical guarantees on its estimation accuracy in the binary case. Compared to previous works on 1-bit matrix completion, our method has some important advantages. First, it works under quite mild assumptions on the sampling distribution. Second, it requires only an upper bound on the maximal absolute value of the unknown matrix. Finally, the rates of convergence given by Theorem 2 are faster than the rates of convergence obtained in [8] and [6]. In future work, we could consider the extension to more general data fitting terms, and to possibly generalize the results to tensor formulations, or to penalize directly the nuclear norm of the matrix probabilities themselves.

**Acknowledgments**

Jean Lafond is grateful for fundings from the Direction Générale de l'Armement (DGA) and to the labex LMH through the grant no ANR-11-LABX-0056-LMH in the framework of the "Programme des Investissements d'Avenir". Joseph Salmon acknowledges Chair Machine Learning for Big Data for partial financial support. The authors would also like to thank Alexandre Gramfort for helpful discussions.

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
