[Reviews · NeurIPS 2014]

Submitted by Assigned_Reviewer_21

SUMMARY

This paper proposes a nuclear norm penalized estimator for matrix completion problem, where the observations take a finite (discrete) number of values. Both with theoretical analysis and with numerical experiment, the authors verify the proposed approach is effective.

REVIEW

1. I understand that there are cases where the observations are discrete and that we may need a distinguished algorithm for them, the recommendation systems may not be a good example. Although most recommender system datasets allow finite number of possible ratings (usually 1 to 5 stars), the output does not need to be finite. Instead, real-valued outputs generally perform well in literature as well as in real-world applications. (For example, when we estimate how many stars a user will give to a movie, users feel more accurate when it is 4.52 rather than 4 or 5.)

2. Related to the above, using multinomial model for recommender systems loses lots of important information. The scores 1 to 5 are not actually categorical, but have order: 1 < 2 < 3 < 4 < 5. Thus, incorrect estimation between two some categories may not be equivalent depending on what they are. Estimating 5 as 4, for example, is not equivalent to estimating 5 as 1. This model seems losing this information, and the evaluation evaluating only with the error rate also overlooks this effect. (Please see detailed comment in #4.)

3. The execution time shown in Table 1 is impressive, as I know nuclear norm minimization is in general not fast. Have you tried bigger size than 10K * 10K? It would be nicer if we see 100K * 100K or so, as it is common to take like 24 hours for training a recommendation systems with large-scale dataset. Also, MovieLens have two bigger ones. Please include the execution time for MovieLens 100K, and maybe 1M or 10M as well.

4. The evaluation criteria used for real world data was just binomial prediction error. Assuming that this is the ratio of estimation to an incorrect category, this measure penalizes incorrect by 1 level (4 and 5, 3 and 4, and so on) and incorrect by 2 or higher levels (2 and 5, 1 and 4, etc) equally. They would be completely different, in practice. Can you provide MAE or RMSE score as well? In this way, it is also possible to compare the performance with many other CF algorithms in literature.

5. According to Table 4, rating 3, 4, and 5 shows much higher error rate than rating 1 and 2. Unfortunately, real-world recommendation systems almost do not care about low ratings (< 4), as we are interested only in "preferrable" items for each user. Some papers even evaluate recommendation algorithms using asymmetric measures, penalizing errors in high rating much severely. (For example, see this experimental study: http://arxiv.org/pdf/1205.3193.pdf)

6. Regarding presentation: the shape and meaning of X was not clearly defined. I could understand what is X later, but it would be nicer to define X explicitly and clearly at the beginning of Notations section. Some typos I found:
- In Theorem 1: "Assume 1 holds" --> "Assume H1 holds"
- Last sentence of section 3.2: "is give" --> "is given"
- In section 3.3: "Figure 3.3" and "Table 3.3 and 3.3" --> "Figure 1" and "Table 2 and 3"
- In section 3.4: "For this datasets" --> "For this dataset" or "For these datasets"

AFTER REBUTTAL

Thank you for your response. I see that the proposed framework can take ordinal classes into account. The experimental part is still not scalable, but I agree with the authors' response that this paper conveys mainly theoretical results.
Summary: This paper presents a novel approach for matrix completion problem taking categorical values. The paper may be improved by addressing issues related to experiment and presentation.

Submitted by Assigned_Reviewer_31

This paper proposes a general multinomial logit for matrix completion
with observation on a finite set of values. I think that the paper is
interesting, mathematically correct and quite new. It is correctly
written, although there is some room for improvement. Here are my
comments.

- p.2 l.22: ... knowledge of an upper bound on the absolute values of
the probabilities ... : this is missleading, as a probability os
between 0 and 1, please reformulate

- Maybe first describing the mutinomial logit case gives a better
understanding the model, then describing the more general model

- typo p.3 function$\Pi$

- Please mention what the constants $M_\gamma$, $L_\gamma$ and
$K_\gamma$ are in the Multinomial logit case

- An explanation giving intuition for the considered penalization
could be helpful. For the p=2 case, this is standard trace norm
penalization, for the multiclass problem, it is not intuitively
clear why the parameter matrices separating each class from class p
should be separately low-rank

- p.3 a comment on the reason why results are given only for $p=2$
classes should be given, although I guess this is for technical
reasons

- Assumption H1: we can only have $\mu \geq 1$

- A simpler statement in Theorem~2 or a take home message about the
obtained rate could be helpful, in the simplest set of assumptions
(uniform sampling of entries, square matrix, etc.), for instance a
sentence mentioning that we recover the "usual" matrix completion
rate of convergence of order rank * dimension / (number of entries)
could be helpful

- p.5 l4 $h^j$ stand for canonical basis, not $e^j$

- In Algorithm 1 the for loop for j start at 1, not 0

Summary: A good paper about a multi class matrix completion, with nice theoretical results and a scalable algorithm, with convincing experiments

Submitted by Assigned_Reviewer_46

** FROM AUTHOR FEEDBACK AND DISCUSSION WITH OTHER REVIEWERS **

From discussion with other reviewers, the bounds in this paper are much better than previous results, but the difference and impact are poorly explained:

BOUND AND IMPACT (from discussion)

-- The squared Hellinger is dominated by Kullback, hence a bound with Kullback is stronger than Hellinger...

-- ...the bounds proposed in this paper are really much better than previous results, but this is indeed badly explained in the paper.
The bounds proposed in this paper are of order (assuming square d * d matrix to simplify)

max( r d log d / n, r (log d / n) ^(1/2) )

where r = rank, d = number of rows = number of columns and n number of observed entries

while previous bounds from [6] and [8] are of order

r (d / n) ^ (1/2)

Recalling that in matrix completion problem one typically has n >> d (otherwise no recovery is possible), this is a real improvement. Actually the result from this paper matches the results from Koltchinskii Tsybakov and Lounici (2011), that gives the best generalization errors for matrix completion to be found in literature.

-- one doesn't need to know an upper bound for the rank, while [6] does.

SUMMARY:

Quality score updated to 6 due to novel theoretical contribution, after consultation with other reviewer (above).

If the practical benchmark wasn't lacking (there are many standard baselines for MovieLens which are ignored), quality would be 7.

If the paper discussed and illustrated the practical significance of T2, quality would be 8.

------------------------------------------------
MAIN IDEAS OF THE PAPER

The paper addresses the reconstruction of a matrix based on partial observations. The entries are discrete. It is assumed that the discrete entries are generated from a likelihood function f(X) that depends on matrix X (for the binary case) or p-1 matrices X_p for p categories. It is done by maximizing the likelihood over the observed entries (the same entry index can be observed more tha once, according to some \Pi distribution), but regularizing the model with a nuclear or trace norm constraint on X.

As the true X is unknown, we would like to know how far off X^ is, in terms of the Kullback-Leibler (KL) divergence between f(X) and f(X^). For this two upper bounds are given. A version of the Rank 1 Descent (R1D) algorithm is given to solve for X^.

The bounds are for 1-bit (p=2); experimental results are for the general case.

The experiments are on a simulated data set, and MovieLens100K. In it, the paper shows that when the data is sampled from a low-rank model with logistic likelihood, the algorithm better recovers the true sampling probabilities than one that assumes a Gaussian likelihood instead of a logistic one. On MovieLens100K, which has ratings 1 to 5, the logistic model has a smaller predictive error than a Gaussian model.

QUALITY / CLARITY

The paper seems to stand on two disconnected legs. An upper bound on the KL divergence between f(X) and f(X^) is provided in Theorem 2 for 1-bit matrix completion, but the paper does not explain or show its practical significance. It might be informative if the experiments have bearing on the developed theory, for instance, by including the KL bounds from T2 in Figure 1. Understandably, the bound relies on unknown constants, in which case a relevant interpretation of the bound is required.

For these reasons I’m not sure the reader is adequately informed. After the main bound is stated, it is never stated, discussed, or evaluated.

ORIGINALITY

The paper addresses a low-rank matrix completion for categorical random variables, which is a new contribution.

There is a marginal extension of R1D (Algorithm 1).

SIGNIFICANCE

It is not clear that the state of the art is advanced in a demonstrable way. There is a copious amount of literature evaluating 1 to 5 star rating prediction on the MovieLens and Netflix data sets, all of which are evaluated baselines that can be copied verbatim from known work.

ADDITIONAL COMMENTS / SUGGESTIONS

The conclusion states that the rates of convergence in T2 are faster than that of Davenport and Cai's. I did not read their papers (Davenport's arxiv.org/abs/1209.3672) and Cai and Zhou (JMLR 2013) in extreme detail, but it seems that their bounds are in terms of Hellinger distance and Frobenius norm? It would be nice to know from the paper how the rates are faster. If there is a rebuttal period, will the authors please add this information? And clearly demonstrate how the bound advances the state of the art, or gives new insights?

The binary case has been studied before outside the “1 bit” setting, for instance in [Matchbox: Large Scale Bayesian Recommendations; Stern, Herbrich, Graepel, in WWW 2009]. The likelihoods are similar to that employed in the paper (logit, probit), although the rank is explicitly constrained.

Would the likelihood in the baseline evaluation (line 315) be the author's model (replacing p with X), except that a probit instead of logit link function is employed?

For discrete ordinal ratings, a stochastic ordering is assumed in the input space, so that for four categories “awesome” > “good” > “mediocre” > “bad” are stochastically ordered [Regression models for ordinal data; McCullagh, in Journal of the Royal Statistical Society B, 1980]. It is a property that the model does not capture, but one that is very applicable to the MovieLens100K evaluation. See for instance [A Hierarchical Model for Ordinal Matrix Factorization; Paquet, Thomson, Winther, in Statistics and Computing, 2012] for ordered discrete alphabet and movie ratings.

The paper could demarcate where / how R1D is extended.

Lines 192 and 358: T2 suggests a setting for lambda.

Line 410: Offsetting or shifting with a bias is common in the “Gaussian model” in literature: why not mean-centre the ratings? Analogously, it’s done for Principal Component Analysis.

Typos, etc.
Line 90: Shatten q-norm defined, never used (except q=1).
Line 98: Hellinger distance defined, never used.
Sections 3.1 and 3.2. Notation between e^j and h^j is switched around Line 220?
Lines 219 and 250: Multiple definitions
Line 355: Figure 3.3
Line 360: Table 3.3 and 3.3

Summary: The paper introduces a low-rank matrix completion for categorical random variables (new), with a KL-bound. The bounds in the paper are better than previous results, although the practical significance of the bound is left unattended in the evaluation.
Author Feedback
Author rebuttal: First, we would like to thanks the reviewers for their valuables comments and interesting references. The various typos pointed out will of course be removed in the camera-ready paper upon acceptation, and the references cited appropriately.

Then, we would like to emphasize that this paper is mainly considering theoretical results for matrix completion, and is not proposing a new state-of-the-art algorithm for recommender systems. Our main focus is on matrix completion from discrete observations. This kind of data was considered in practice as an illustration of our theoretical and practical results, and show that common datasets could be dealt with using the implementation of our proposed method.

ORDINAL RATINGS: The reviewers have mentioned that our method does not handle the case of "ordered" classes. This is true for the experiments we performed with the logit link function. Nevertheless, our framework is general enough to allow each f^j to depend on j. So, using for instance the probit link function, one can easily take the ordering into account. This is of course a nice suggestion, which could be dealt with within our framework. One should also note that for some datasets, the observations could be purely qualitative (e.g., colors for clothes).

A major criticism stated that multinomial models were not
relevant for recommendation system, since only the 5 stars are of interest. Note that as a particular case, this could be taken into account using the simple "5 vs all" binomial approach handled by our work.

INTUITION: Comments by Reviewer_31: "An explanation giving intuition for the considered penalization could be helpful"

For the case p=2, our choice is the classical low rank assumption on the parameter X. For p>2, our motivation is that the distribution associated to the i-th user and j-th item can be summarized by a combination of a few elementary distributions. Each elementary distribution is parametrized by a vector whose components are element-wise products of (user term) \times (item term).

THEORY: We want to stress out the following important
theoretical advances:
-we can recover the "usual" matrix completion rate of convergence; that was not the case in the previous works;
-we can deal with possibly non-uniform sampling;
-our method is adaptive, no prior bound on the rank is required.

Comments by Reviewer_31: "comment on the reason why results are given only for $p=2$ classes should be given":
Currently, we can prove minimax optimal rates only for p=2 with our proof techniques. The case p>2 can be handled under a separability assumption of the likelihood, or to the price of optimality (in the minimax sense).

The following comments on the main theorems will be added after their statement, as proposed by two reviewers:

Note that, up to the factor $L_\gamma^2/K^{2}_\gamma$,
the rate of convergence given by Th. 2, lead to the same as in the case of usual unquantized matrix completion, see, for example, \cite{Klopp14} and \cite{Koltchinskii_Tsybakov_Lounici11}. For this usual matrix completion setting, it has been shown in \cite[Theorem 3]{Koltchinskii_Tsybakov_Lounici11} that
this rate is minimax optimal up to a logarithmic factor.
Let us compare this rate of convergence with those obtained in previous works on 1-bit matrix completion.
In \cite{Davenport_Plan_VandenBerg_Wootters12}, the parameter X$ is estimated by minimizing the negative log-likelihood under the constraints
$\|\mat{X}\|_\infty \leq \gamma$ and $\|\mat{X}\|_{\sigma,1} \leq \gamma \sqrt{r m_1 m_2}$
for some $r>0$. Under the assumption that $\rank(\mat{\tX}) \leq r$, they could prove that
\begin{equation*}
\frac{\|\mat{\tX}-\hat{\mat{X}}\|^2_{\sigma,2}}{m_1 m_2} \leq C_\gamma \sqrt{\frac{rd}{n}},
\end{equation*}
where $C_\gamma$ is a constant depending on $\gamma$
(see \cite[Theorem 1]{Davenport_Plan_VandenBerg_Wootters12}). This rate of convergence is slower than the rate of convergence given by Th.2. \cite{Cai_Zhou14} studied a max-norm constrained maximum likelihood estimate and obtain a rate of convergence similar to \cite{Davenport_Plan_VandenBerg_Wootters12}.

Note also that more details are provided in the supplementary material (see Lemma 1 for instance to connect Hellinger and KL).

ALGORITHM: Our algorithm extends "Rank 1 Descent (R1D)" in the fact that it deals with a p-dimensional parameter (tensor), what was not considered in Dudik et al. However, both are identical when p=2. As asked by Reviewer_21 we have now added the computing time for the
MovieLens 10M. It took 940s to perform.

In the experiments, for fair comparisons, the \lambda parameters were all chosen by cross-validation, over geometric grid, with the same type of upper bounds as for the LASSO (cf. scikit-learn implementation of LassCV for instance).

PERFORMANCE CRITERION:
Comments by Reviewer_21: "Can you provide MAE or RMSE score as well? In this way, it is also possible to compare the performance with many other CF algorithms in literature". We have performed our experiments and measured prediction error (i.e. the average number of mistakes made). The criterions wanted would require to launch new experiments since the cross-validation should be performed on the same criterion as well. This will be done for the camera ready paper.